# Prion infection, transmission, and cytopathology modeled in a low-biohazard human cell line

Merve Avar[1,*], Daniel Heinzer[1,*], Nicolas Steinke[1], Berre Doğançay[1], Rita Moos[1], Severine Lugan[2], Claudia Cosenza[2], Simone Hornemann[1], Olivier Andréoletti[2], Adriano Aguzzi[1]

Transmission of prion infectivity to susceptible murine cell lines has simplified prion titration assays and has greatly reduced the need for animal experimentation. However, murine cell models suffer from technical and biological constraints. Human cell lines might be more useful, but they are much more biohazardous and are often poorly infectible. Here, we describe the human clonal cell line hovS, which lacks the human *PRNP* gene and expresses instead the ovine *PRNP* VRQ allele. HovS cells were highly susceptible to the PG127 strain of sheep-derived murine prions, reaching up to 90% infected cells in any given culture and were maintained in a continuous infected state for at least 14 passages. Infected hovS cells produced proteinase K–resistant prion protein (PrP^Sc), pelletable PrP aggregates, and bona fide infectious prions capable of infecting further generations of naïve hovS cells and mice expressing the VRQ allelic variant of ovine PrP^C. Infection in hovS led to prominent cytopathic vacuolation akin to the spongiform changes observed in individuals suffering from prion diseases. In addition to expanding the toolbox for prion research to human experimental genetics, the hovS cell line provides a human-derived system that does not require human prions. Hence, the manipulation of scrapie-infected hovS cells may present fewer biosafety hazards than that of genuine human prions.

## Introduction

Prions, the causative agent of transmissible spongiform encephalopathies, are devoid of nucleic acids and consist primarily of a protein termed PrP^Sc. These characteristics differentiate prions from viruses and have profound consequences on the methodologies applicable to their study. Viral replication can be assessed by quantifying the viral nucleic acids, but this is not possible for prions. Moreover, PrP^Sc cannot be reliably distinguished from its cellular precursor PrP^C in living cells, making it impossible to assess prion replication in real time. Finally, the study of human prions is fraught with serious biosafety concerns because prion contaminations of laboratory equipment are difficult to detect, prions are exceedingly sturdy and difficult to inactivate, and there are neither vaccines nor therapies against prion infections (Taylor, 1999; WHO, 2000; Leunda et al, 2013; Aguzzi et al, 2018).

Despite the above obstacles, cellular models of human prion replication and toxicity are crucial to advancing our understanding of human prion diseases. Cell culture models of prion infections have enabled the discovery of certain molecular players responsible for prion infection and propagation. However, most of the in vitro models are based on mouse cell lines such as N2a subclone PK1 (Klöhn et al, 2003), CAD5, and GT-1/7 (Solassol et al, 2003), which may not reproduce all characteristics of human prions. Most importantly, with few exceptions (Schätzl et al, 1997), the infection of these cell lines with prions does not result in a measurable pathological phenotype, a finding that limits their usefulness for disease research.

Currently, there are only three reports of human cellular models for prion infection and propagation (Ladogana et al, 1995; Krejciova et al, 2017; Groveman et al, 2019). However, the culture and maintenance of these models are costly, extremely laborious and have limited scalability. Finally, a major limitation of the above models is that human prions derived from postmortem brain matter from patients succumbing to Creutzfeldt–Jakob disease (CJD) must be used as inoculum. This raises bioethical issues, requires the availability of a biosafety level three (BSL3) facility, which restricts the usage to only a few laboratories worldwide, and exposes laboratory workers to potential risks of infection. For all these reasons, the lack of broadly applicable human cell culture models for prion diseases has been a limiting factor in the understanding of the mechanisms behind the formation, propagation, clearance, and toxicity of prions.

We reasoned that the problem of biosafety may be attenuated through the use of gene replacement. Ovine prions, which cause sheep scrapie, have not been reported to cause prion diseases in humans. Although scrapie is endemic in many sheep flocks (Detwiler & Baylis, 2003; Houston & Andréoletti, 2019) and sheep brain and

---

[1]Institute of Neuropathology, University of Zurich, Zurich, Switzerland    [2]UMR INRA/ENVT 1225 IHAP, École Nationale Vétérinaire de Toulouse (ENVT), Toulouse, France

Correspondence: adriano.aguzzi@usz.ch
*Merve Avar and Daniel Heinzer contributed equally to this work

spinal cord are considered fit for human consumption (EFSA Panel on Biological Hazards, 2015) in many countries, there is no epidemiological evidence connecting the latter with CJD (Brown et al, 1987; van Duijn et al, 1998; Georgsson et al, 2008). Transmission of scrapie to mice expressing human PrP$^C$ was attempted, but ovine prions arising from VRQ allelic variant sheep have failed to transmit disease efficiently and mice succumbed to disease only in the second passage (Cassard et al, 2014). Although these data do not conclusively prove that sheep prions are innocuous to humans, they suggest that the handling of ovine PrP$^{Sc}$ in a laboratory setting may be less dangerous than the manipulation of human prions. Hence, the replacement of the human *PRNP* gene with its ovine counterpart may lead to a cell line that retains all characteristics of human cells, while lowering potential biohazards.

Here, we used the human neuroblastoma cell line, SH-SY5Y (Pease et al, 2019), with a deletion of the human *PRNP* gene (SH-SY5Y$^{ΔPRNP}$) and inserted instead the *Ovis aries PRNP* gene (V136-R154-Q171 [VRQ] variant). We report that the resulting clones expressed the ovine PrP$^C$ and were infectible with the ovine prion strain PG127 (Andréoletti et al, 2011). Cells remained permissive for prion propagation through multiple passages and exhibited characteristic prion-induced cytopathic effects.

# Results

## Expression of ovine PrP$^C$ in ovSH-SY5Y cells

The VRQ allelic variant of sheep PrP$^C$ was reported to convey high susceptibility for ovine prions to xenogenetic cell lines such as the rabbit-kidney RK13 cells (Vilette et al, 2001) and even to invertebrates such as *Drosophila melanogaster* (Thackray et al, 2018). To maximize the likelihood of obtaining a prion infection–permissive cell line, we inserted the ovine *PRNP* reading frame with the VRQ genotype under the control of the ubiquitous EF1α promoter in SH-SY5Y$^{ΔPRNP}$. In addition, the human secretory signal peptide of PrP$^C$ was added to the ovine *PRNP* sequence to facilitate proper biogenesis and targeting to the secretory pathway. Stably transfected cells were kept under G418 selection and expanded as a polyclonal bulk (Fig 1A).

For further experimentation, we isolated a polyclonal cell line, hereafter referred to as "povS" (polyclonal ovine PrP$^C$-expressing SH-SY5Y$^{ΔPRNP}$) and a monoclonal cell line obtained through limiting dilution and referred to as "hovS" (monoclonal ovine PrP$^C$-expressing SH-SY5Y$^{ΔPRNP}$). To characterize the two cell lines, we used Western blotting and immunocytochemistry. Western blotting showed that the transgenic, ovine PrP$^C$ in both cell lines was expressed at comparable levels as in the two human cell lines, LN229 and U251-MG, and at higher levels as in wild-type (wt) SH-SY5Y cells (Fig 1B) (Pease et al, 2019). As expected, no expression of PrP$^C$ was detectable in SH-SY5Y$^{ΔPRNP}$. In addition, immunocytochemistry showed an equally distributed staining for ovine PrP$^C$ at the cell surface of the hovS, confirming that PrP$^C$ was localized at the cell membrane, whereas the distribution of PrP$^C$ for the polyclonal povS was more heterogeneous (Fig 1C). These data indicate that both cell lines express sufficient amounts of cell surface exposed PrP$^C$ and may, thus, allow for efficient replication of prions.

## Propagation of sheep prions in ovSH-SY5Y cells

We next investigated the permissiveness of hovS and povS and their capability to replicate sheep prions. We, therefore, infected the cells (3 × 10$^5$ cells/six well) with 18.75 μl/well of a 20% brain homogenate from PG127 prion-infected tg338 mice expressing the ovine PrP$^C$ (allelic variant VRQ–VRQ) (Andréoletti et al, 2011). Cells treated with noninfectious brain homogenate (NBH) from C57BL/6J mice were used as control. After 3 d, the medium was aspirated and cells were passaged numerous times. Several passages were tested for the presence of proteinase K (PK)–resistant PrP (PrP$^{Sc}$). At passage 8 (used here and henceforth, unless otherwise stated), PrP$^{Sc}$ became reliably detectable in the monoclonal hovS as indicated by PK Western blotting (2.5 μg/ml PK for 50 μg total protein) but not in the polyclonal povS (Fig 2A and B). Interestingly, the glycosylation pattern and the electrophoretic mobility shifts of PrP$^{Sc}$ in hovS differed from those of the original PG127 inoculum and were more reminiscent to those of the previously reported prion-propagating ovinized RK13 cells (Vilette et al, 2001). Moreover, the diachronic PrP$^C$ expression pattern observed on non–PK-digested immunoblots showed already increased amounts of PrP post infection, which we hypothesized to be due to the high levels of PrP$^{Sc}$ in the ovinized cells.

A hallmark of productive prion infection is the aggregation of misfolded PrP into higher order structures. We, therefore, set out to investigate whether such structures might be present in PG127-infected ovinized SH-SY5Y cells. Cell lysates were separated into supernatant and pelleted fractions and analyzed by immunoblotting with and without PK digestion using anti-PrP antibody POM1 (Polymenidou et al, 2008) for detection. Although a comparable PrP$^C$ signal was present in the NBH-treated hovS in both fractions, we observed a stronger signal in the concentrated fraction of the PG127-infected hovS and povS cells (Fig 2C). We conclude that the infected cells indeed formed higher molecular weight PrP entities upon prion infection. In addition, upon PK digestion, PrP was detected only in the concentrated fraction for both povS and hovS, leading to the conclusion that the detected PrP$^{Sc}$ in the infected cells was associated with aggregates. The lack of PrP$^{Sc}$ in the knockout cells indicated that it was produced de novo by the ovinized cells and was not due to residual signal arising from the original inoculum. To investigate for the presence of PK-sensitive PrP species and to further examine differences in biochemical properties, a pronase E digestion and Western blotting (45 min, 37°C, 200 μg/ml for brain homogenate and 20 μg/ml for cell lysate) was performed with lysate of PG127-infected hovS and the original PG127 inoculum. Again, a differential glycosylation pattern and differences in the electrophoretic mobility were detected for PG127-infected hovS and the inoculum (Fig 2D).

## Detection of infected cells at single-cell resolution

To determine the percentage of infected cells, ELISpot assays were performed (Arellano-Anaya et al, 2011). Three decadic dilutions of prion-infected hovS and povS (40,000, 4,000, and 400 cells) and NBH-treated and PG127-infected SH-SY5Y$^{ΔPRNP}$ controls from the same passage were spotted and digested with PK, followed by denaturation with guanidinium thiocyanate and detection with

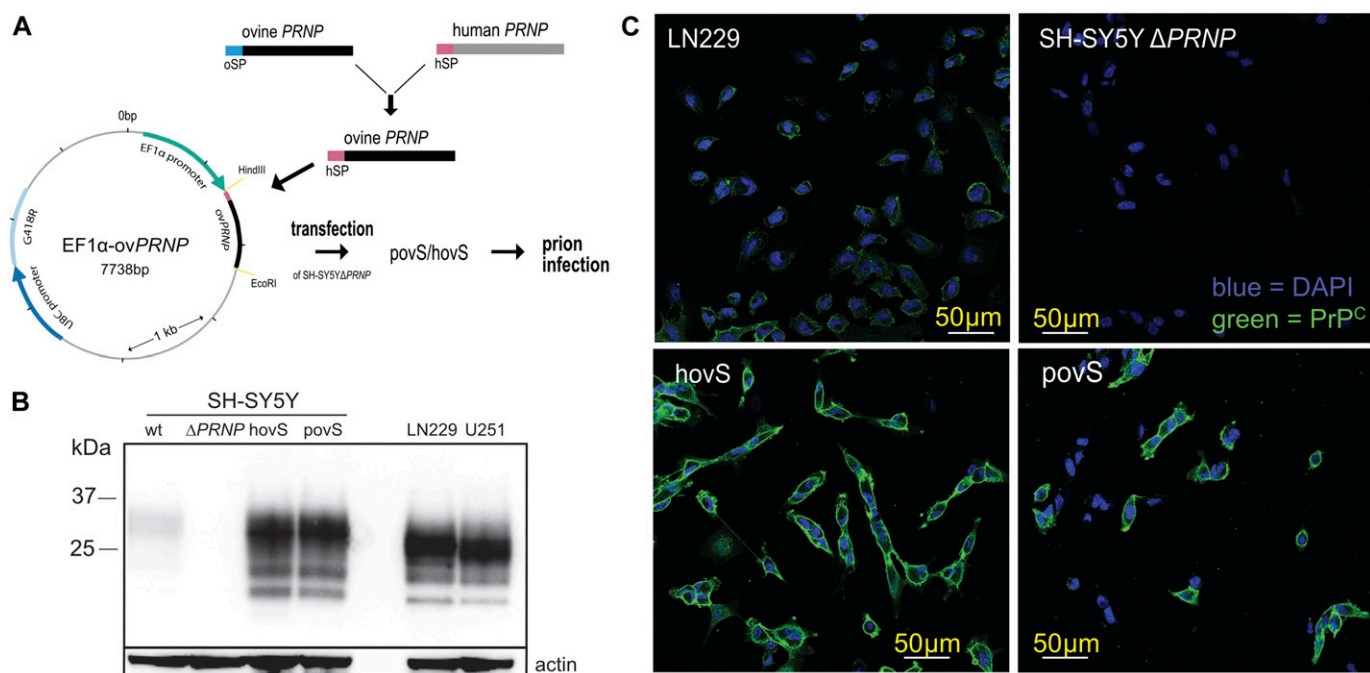

**Figure 1. Generation and characterization of the ovinized SH-SY5Y cell lines.**
**(A)** Generation of SH-SY5Y$^{\Delta PRNP}$ cell lines expressing the ovine VRQ PrP$^C$ variant and its subsequent infection with the PG127 strain of sheep-derived prions passaged in tg338 mice. *Hin*dIII and *Eco*RI restriction sites were used to clone the ovine *PRNP* construct. Ticks in the plasmid map correspond to increments of 1,000 base pairs. hSP, human signal peptide (purple); oSP, ovine signal peptide (blue); hovS, monoclonal ovSH-SY5Y; povS, polyclonal ovSH-SY5Y; ovSH-SY5Y, SH-SY5Y$^{\Delta PRNP}$ transfected with a plasmid harboring the sequence for ovine *PRNP* (ov*PRNP*). **(B)** Western blot analysis comparing the expression levels of PrP$^C$ in hovS and povS with those in wt SH-SY5Y and in the human cell lines U251-MG and LN229. HovS and povS showed similar PrP$^C$ expression levels as U251-MG and LN229, whereas the levels in wt SH-SY5Y were slightly lower. SH-SY5Y$^{\Delta PRNP}$ cells were used as negative control and actin as loading control. The anti-PrP antibody POM2 was used for detection. **(C)** Confocal imaging to detect cell surface exposed PrP$^C$ on hovS and povS. hovS and a subpopulation of povS showed a strong signal for cell surface exposed PrP$^C$, whereas no detectable signal was visible for SH-SY5Y$^{\Delta PRNP}$. LN229 cells were used as positive control. The anti-PrP antibody POM1 (here and henceforth) was used for detection of PrP. Source data are available for this figure.

POM1. The percentages of infected cells were found to be 86% for hovS and 12% for the povS polyclone (n = 3 wells of 400 cells/well) (Fig 2E). NBH-treated cells and prion-infected SH-SY5Y$^{\Delta PRNP}$ cells remained free of spots at the lowest dilution. We conclude that ovSH-SY5Y cells were permissive for prion infection with a high infection rate and that the number of PrP$^{Sc}$-positive cells was dependent on the number of PrP$^C$-expressing cells.

### Cytopathic effects in infected ovSH-SY5Y

Prion propagation in cell culture has been documented for well over five decades (Clarke & Haig, 1970; Solassol et al, 2003). However, in most instances, prion replication does not appear to induce any cytopathic phenotype, except of vacuolation seen in GT-1/7 cells, a mouse GnRH-positive cell line of hypothalamic origin infectible with the Rocky Mountain Laboratory strain of prions (Schätzl et al, 1997). Surprisingly, a vacuolation phenotype (Fig 2F) in infected hovS cells became evident after three passages post inoculation. Vacuolation increased steadily as cells were kept longer in culture and the phenotype was especially prominent upon splitting. It lessened as the cells grew to confluency and increased in a cyclic manner upon new passaging. This phenomenon was not observed in NBH-treated cells or in SH-SY5Y$^{\Delta PRNP}$

cells inoculated with PG127 prions. In addition, the polyclonal line, povS, did not show an appreciable amount of vacuolation, potentially because of a lower proportion of PrP$^C$-expressing cells not allowing for large amounts of prion propagation. Infected hovS also displayed impaired cell growth when compared with NBH-treated hovS (Fig 2G). This further indicates that prion replication led to a dysregulation in cellular metabolic processes.

### Investigation of seeding properties of prion-infected ovSH-SY5Y cells

To investigate whether prion-infected ovSH-SY5Y cells exhibited self-propagating activities, lysates of cells from passage 8 were subjected to real-time quaking induced conversion (RT-QuIC) using two different dilutions (1:50 and 1:250) (Atarashi et al, 2011; Frontzek et al, 2016). Both cell lines induced an increase in the thioflavin T (ThT) signal over 105 h, whereas the signals for the lysates of NBH-treated cells and of SH-SY5Y$^{\Delta PRNP}$ remained negative (Fig 3A). This indicates that prions produced by the infected cells are in fact capable to seed PrP$^C$ into an aggregated form. Intriguingly, povS also showed a delayed ThT signal, in contrast to showing no signal upon PK-Western blotting (Fig 2B).

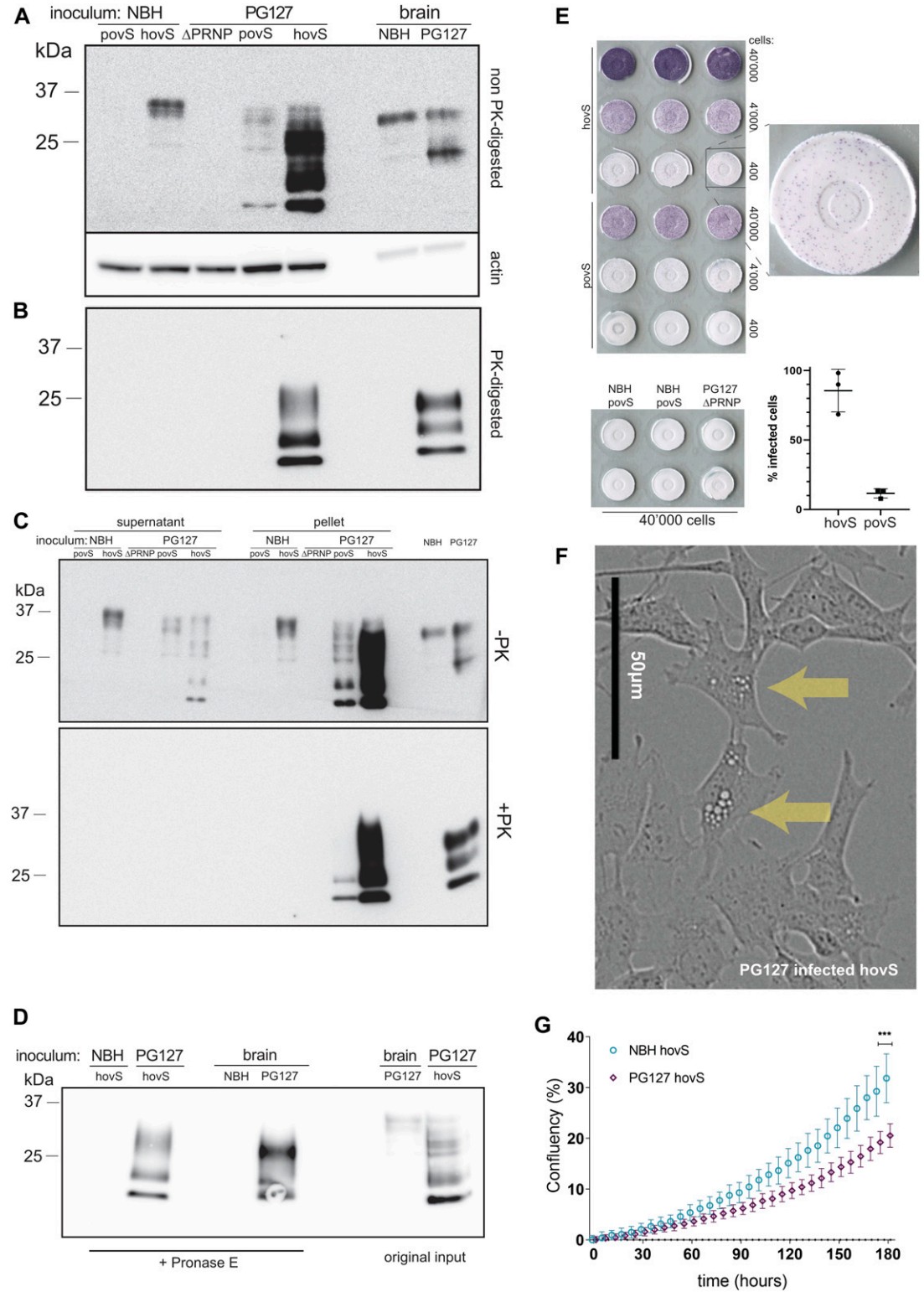

**Figure 2.    PG127-infected povS and hovS show altered electrophoretic profiles, formation of protease-resistant PrP^Sc, and enhanced cytopathic effects.**
**(A)** Western blot analysis of PG127-infected povS and hovS cells indicated a different glycosylation pattern, a shift in the electrophoretic mobility, and partially protease-resistant PrP^Sc, when compared with noninfectious brain homogenate (NBH) exposed–hovS. Rightmost lanes: NBH- and PG127-infected brain homogenate. PG127-infected SH-SY5Y^ΔPRNP lysate was used as negative control. -PK = non–PK-digested. **(A, B)** Same samples as in (A), but digested with proteinase K (PK). PrP^Sc is visible in PG127-infected hovS and the PG127 inoculum. No detectable bands are visible for PG127-infected povS, possibly because of the lower expression of PrP^C in these cells. **(C)** Western blot analysis of pellets and supernatants of cell lysates. The aggregate-enriched pellets of PG127-infected povS and hovS displayed a stronger signal for both

## Lysates from infected cells transmit infectivity to newly inoculated cells and tg338 mice

We then sought out to determine whether the ovinized cells are also able to propagate bona fide prions, herewith defined as microbiologically active infectious agents. To assess whether the newly formed prions could convey infectivity, we inoculated freshly cultured hovS with lysates of PG127-infected hovS cells, that had revealed positive seeding activities in the RT-QuIC, for 3 d. Cells were passaged eight times to dilute out the initial inoculum and to allow for prion propagation. PK-Western blotting revealed that lysates of ovinized cells were capable of conveying infectivity to freshly cultured cells. These results suggest that ovSH-SY5Y produce bona fide prions (Fig 3B). Moreover, the vacuolation phenotype was also evident in the hovS infected with the lysate of PG127-infected hovS, which is in line with the observation that these cells are permissive for persistent prion infection. To ultimately confirm the bona fide nature of prions produced by the ovinized SH-SY5Y, PG127-infected hovS and SH-SY5Y$^{\Delta PRNP}$ as well as NBH-treated lysates, each harvested in PBS from one well of a six-well plate, were used to inoculate tg338 mice (n = 6). All mice inoculated with lysates of infected hovS succumbed to disease after 72 ± 2.5 d, whereas mice inoculated with the control lysates are viable and show no sign of disease 130 days post inoculation (Fig 3C).

## Determining seeding capacities for different substrates

The strain properties of prions are encoded within the inoculum (Aguzzi & Weissmann, 1997). However, the genetic makeup of the host and environmental factors can shift the properties of prion strains. We, therefore, examined the seeding properties of prions formed by hovS by measuring their efficiency in seeding substrates from different species. Lysates of PG127-infected hovS, SH-SY5Y$^{\Delta PRNP}$, or NBH-treated cells were applied to the protein misfolding amplification assay (PMCA) (Lacroux et al, 2014; Douet et al, 2017) using brain homogenates from tg338 (ovine VRQ PrP variant), tgARQ (ovine ARQ PrP variant), tgBov (bovine PrP), tg650 (Methionine 129 human PrP variant), and tg361 (Valine 129 human PrP variant) mice as substrates. Lysates of PG127-infected hovS only seeded prion formation with the ovine PrP$^C$ sequence–containing substrates, tg338 and tgARQ, but not with the bovine PrP$^C$ and human PrP$^C$–sequence containing substrates (Fig 3D). No seeding activity was detected for the controls, PG127-infected SH-SY5Y$^{\Delta PRNP}$, and NBH-treated cells with any of the substrates. These results imply that the seeding properties of the prions formed by the hovS resemble the original PG127 ovine prions and

do not have the propensity to cross the species barrier, despite being produced in a human cell line.

# Discussion

The VRQ variant of ovine PrP has been reported to convey prion infectibility to a wide range of hosts (Vilette et al, 2001; Archer et al, 2004; Thackray et al, 2018). Here, we constructed SH-SY5Y cell lines from which we removed the human *PRNP* gene, instead expressing the VRQ variant of ovine PrP$^C$ under transcriptional control of the housekeeping EF1$\alpha$ promoter. The ovine ER localization signal was swapped with the human sequence, allowing for efficient translocation of the transgenic PrP$^C$ to the cell surface, where the initial contact between PrP$^C$ and PrP$^{Sc}$ is posited to occur (Goold et al, 2011).

After inoculation with the PG127 prion strain and upon serial passaging, cells accumulated conspicuous PK-resistant aggregates. In addition, PG127-infected hovS cells showed cytosolic vacuolation that continued to steadily increase in subsequent passages. The vacuolation was less conspicuous in persistently infected povS cells, maybe because vacuolation is dependent on the concentration of prions. Collectively, these findings hint at the fact that PrP$^{Sc}$-producing cells are not selected against and that mammalian cells possess a machinery to cope with prions to some extent.

Although protease resistance, aggregation and misfolding are considered proxies of prion generation, the true essence of the prion is its capability to infect organisms and self-propagate therein. The data provided here show that hovS cells produce bona fide prions. These data reinforce the observation that the VRQ variant of the ovine PrP$^C$ protein supports prion propagation in disparate genetic backgrounds (Vilette et al, 2001; Archer et al, 2004; Thackray et al, 2018). We found that lysates of infected cells functioned as seeds in both the RT-QuIC and the PMCA reactions. Furthermore, such lysates were capable of infecting batches of naive hovS cells, which then accumulated PrP$^{Sc}$ and acquired a vacuolation phenotype. Finally and most importantly, lysates of infected hovS cells were found to induce scrapie in tg338 mice with an attack rate of 100%, confirming that these cells were producing bona fide prions.

Cellular models may prove essential to understand the pathways of the rapidly progressing neurodegeneration that follows prion replication in the mammalian brain. However, none of the prion-permissive cell lines develop any cytopathological phenotypes in response to prion infection, with the exception of GT-1/7 cells which show mild vacuolation. In addition, one common limitation

total PrP and PrP$^{Sc}$. PG127-infected SH-SY5Y$^{\Delta PRNP}$, NBH, and the original PG127 inoculate were used as controls. **(D)** Western blot analysis of pronase E–digested PG127-infected hovS, to investigate the presence of PK-sensitive PrP$^{Sc}$. PG127-infected hovS differed again in their protease resistance pattern from those of the original PG127 inoculum. NBH, non-digested PG127 inoculum and lysate of non-digested PG127-infected hovS were used as controls. **(E)** ELISpot assay of PG127-infected hovS and povS, visualizing cells harboring PrP$^{Sc}$. Membranes were exposed to decadic dilutions of PG127-infected hovS and povS cell suspensions, PK-digested, and stained with POM1. Positive cells were counted on membranes with 400 cells, as higher cell numbers led to signal saturation in hovS cells. Quantification of positive spots (three replicates) revealed 86% ± 12.5% of infected cells for hovS, and 12% ± 2.7% for povS. No positive spots were detected for NBH-treated hovS and povS and PG127-infected SH-SY5Y$^{\Delta PRNP}$ at 40,000 cells. Data in the graph represent the mean ± SD. **(F)** Phase-contrast image of PG127-infected hovS showing intracellular accumulation of vacuoles (arrows). Scale bar = 50 $\mu$m. **(G)** PG127-infected hovS showed a slower growth rate than NBH-treated hovS over 180 h in culture. Live images were recorded from n = 6 wells for each condition. ***$P$ = 0.0004 ($t$ test at final time point).
Source data are available for this figure.

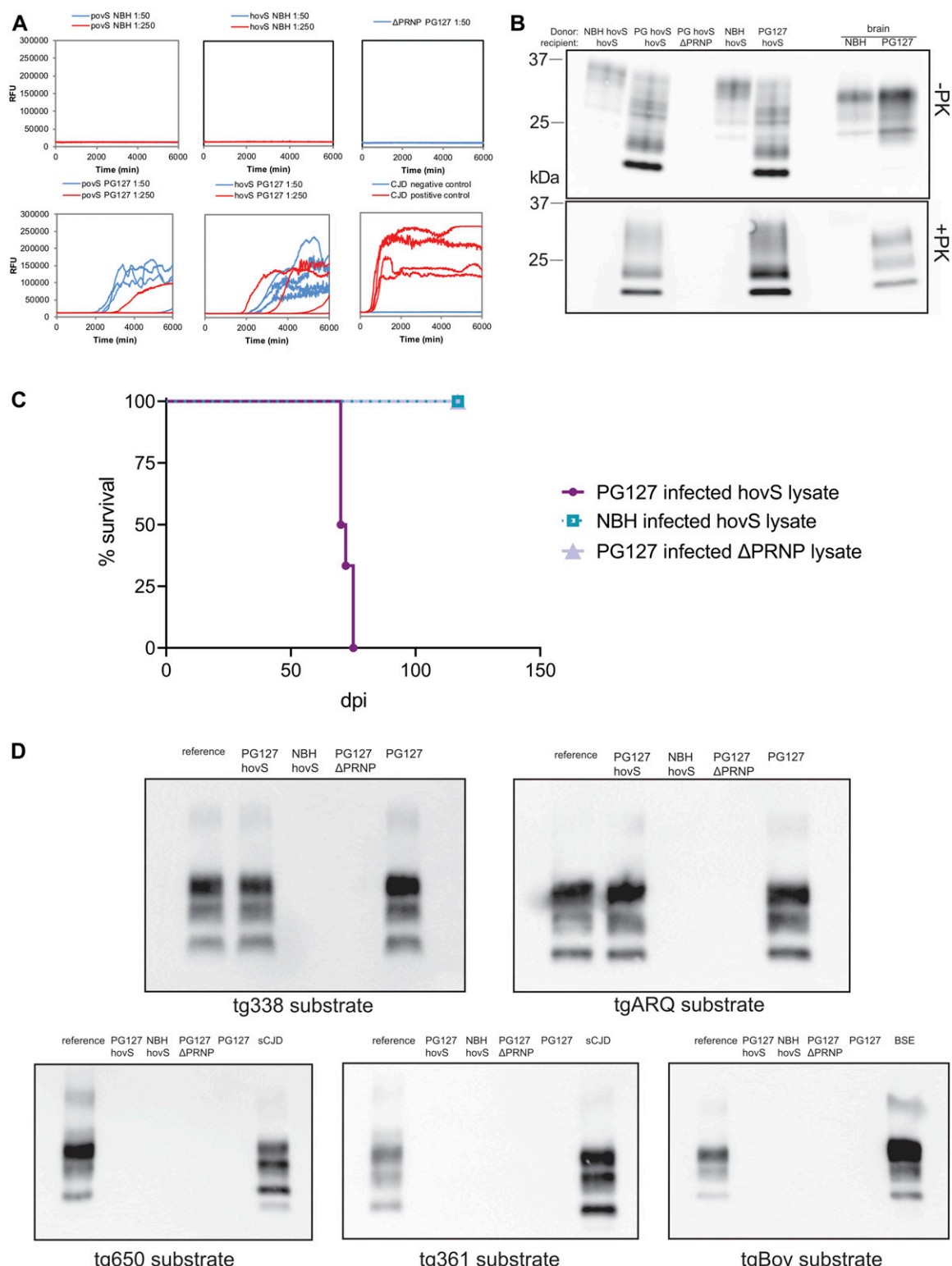

**Figure 3. Self-propagating activity and transmissibility of PG127-infected hovS and povS.**
**(A)** The seeding activity of PG127-infected or noninfectious brain homogenate (NBH)–treated hovS, povS, and SH-SY5Y$^{\Delta PRNP}$ lysates was assessed by RT-QuIC (diluted 1:50 and 1:250). PG127-infected hovS, and to a lesser extent, povS cells induced de novo PrP aggregate formation at both dilutions, whereas cell lysates of either NBH-treated or PG127-infected SH-SY5Y$^{\Delta PRNP}$ used as negative controls did not yield a positive signal. Creutzfeldt–Jakob disease (CJD) and non-CJD brain homogenates were used as positive and negative controls for the amplification reaction. Samples were analyzed in quadruplicates. **(B)** Western blot analysis of serial transmissibility. PG127-infected hovS cell lysates (PG hovS) were used to transmit prion infectivity to fresh hovS cultures. Lysates of undigested and proteinase K–digested hovS exposed to

of prion-infectible cell models is the low percentage of infected cells within each cultured batch (Race et al, 1988; Bosque & Prusiner, 2000). Here, we report that the percentage of prion-infected hovS cells was near 90 percent. The high infection rate and the vacuolation phenotype of hovS cells suggest that the rate of prion formation outcompetes the rate of prion clearance in this clone (Krauss & Vorberg, 2013). This is also in line with our observation of a slower growth rate of these cells in comparison to NBH-treated cultures. The high infection rate of hovS cells and the conspicuous prion-induced cytopathic effects may be related to the ectopic overexpression of PrP$^C$, giving rise to higher prion amounts, which is in contrast to other prion cell models relying on endogenous PrP$^C$ expression (Vilette, 2008). Overexpression of PrP$^C$ did not appear to cause any cytopathology in the model at hand. Therefore, this strategy was adopted instead of CRISPR-mediated gene replacement. An important argument in favor of ectopic overexpression as opposed to gene replacement is that SH-SY5Y cells are aneuploid and have low endogenous PrP$^C$ expression (Fig 1B) which may be insufficient to allow for prion propagation (Yusuf et al, 2013).

Although SH-SY5Y cells express very low levels of endogenous PrP$^C$ (Fig 1B), we elected to inactivate the human *PRNP* before ovination. First, host PrP may be a confounding factor for establishment of infectivity in cell models. Attempts in generation of a relevant human model system for prion propagation may have thus far failed because of expression of endogenous PrP sequestering PrP$^{Sc}$, therefore rendering the cell line impermeable for prion replication (Priola et al, 1994). Second, we strived to minimize the possibility of the cells replicating human prions to protect laboratory workers.

As biosafety is a major concern when working with especially human or bovine prions, we sought to further investigate the possibility of these cells to propagate human prions upon infection with classical scrapie. PK and pronase E digestion showed that the prions formed by ovinized cells had distinct biochemical properties indicative of a strain shift (Aguzzi et al, 2007). These differences were in line with those seen in other cell lines infected with ovine prions and ectopically expressing ovine PrP$^C$ (Vilette et al, 2001; Archer et al, 2004). It has been suggested that not only the strain, but also the tissue responsible for PrP$^{Sc}$ formation can have an influence on the biochemical properties of prions. We used RT-QuIC and PMCA as additional approaches to determine seeding properties of the prions, which was reported to faithfully reproduce strain characteristics (Castilla et al, 2008). Indeed, the prions formed by the ovinized cells could only seed ovine PrP$^C$ as a substrate in PMCA but not human or bovine PrP$^C$. Hence, the biochemical identity of the prions the ovSH-SY5Y cells harbor was different from the original PG127 inoculum, yet the seeding properties did not change upon

passaging in the human cells. Therefore, we suggest that the cellular machinery of the human cell line SH-SY5Y does not impart an ovine-to-human shift onto prions. The prions produced may have a reduced propensity to infect humans. Moreover, hovS cells do not require human material (such as CJD brain) for infection, in contrast to all previously reported human prion cell models (Ladogana et al, 1995; Krejciova et al, 2017; Groveman et al, 2019). For these combined reasons, we posit that hovS cells may be adopted by many research laboratories worldwide without requiring biosafety level three (BSL3) precautions.

As a caveat, however, the zoonotic potential of ovine prions for humans remains controversial. Although the epidemiological data do not suggest the transmission of prions from sheep to humans (Brown et al, 1987; van Duijn et al, 1998; Georgsson et al, 2008), recent studies using ovine prions to inoculate humanized mice (Cassard et al, 2014) and macaques (Comoy et al, 2015) indicate that ovine prions can cross the species barrier—at least under these specific circumstances. However, the penetrance of prion disease in humanized mice was restricted to the second passage, suggesting that it may require strain adaption. In the macaque study, disease progression was reported only in one individual and only after 110 mo with use of a relatively high dose of ovine prions. This is profoundly different from BSE prions which conveyed infectivity to macaques with full penetrance at a much earlier time point and through much lower doses. Considering the WHO recommendation (WHO, 2000) for handling ovine prions in BSL2 facilities and the lack of evidence that ovine prions can transmit infectivity to humans, we suggest that scrapie-infected hovS cells should be handled in BSL2 laboratories with utmost caution.

The present study demonstrates that the ovinized hovS subclone of engineered SH-SY5Y cells is a robust and flexible model of prion replication and cytopathology. Because hovS cells are well characterized, easy to culture, and scalable, they are well suited for high-throughput applications. This opens hitherto unattainable possibilities to address questions pertinent to prion propagation, infectivity, and downstream events in prion pathogenesis in a human cell model which were thus far elusive to the research community.

## Materials and Methods

### Generation of a human monoclonal ovinized SH-SY5Y cell line

For ovinization, a *PRNP* knockout (Δ*PRNP*) SH-SY5Y cell line, described earlier (Pease et al, 2019), was used. *PRNP* coding sequence

PG127-infected hovS lysates displayed the same electrophoretic profiles as the original lysates. Lysates of hovS exposed to NBH-treated hovS and PG127-infected SH-SY5Y$^{ΔPRNP}$ cells were used as negative controls. NBH and PG127 inoculum were loaded as additional controls (rightmost lanes). **(C)** Lysates (20 μl) of PG127-infected hovS and SH-SY5Y$^{ΔPRNP}$ or NBH-treated hovS were intracerebrally inoculated into tg338 mice. All mice succumbed to disease upon inoculation with PG127-infected hovS lysates with an incubation time of 72 ± 2.5 d. Mice inoculated with control lysates do not show any clinical sign of disease >130 (dpi). n = 6 for each condition. **(D)** Lysates of PG127-infected hovS (diluted 1:50) were analyzed for propagation efficiency and substrate specificity by PMCA using substrates from various species and of different genotypes (sheep VRQ/VRQ [tg338], sheep ARQ/ARQ [tgARQ], bovine [tgbov], human 129Met [tg650], and human 129V [tg361]). PMCA reactions of the third round were analyzed for PrP$^{Sc}$ by Western blotting. Lysates of PG127-infected hovS cells showed positive seeding reactions only with the ovine substrates. NBH-treated hovS and PG127-infected SH-SY5Y$^{ΔPRNP}$ were used as negative controls and PG127, BSE, and sCJD prions amplified with the respective substrates as positive controls. Reference: PG127 inoculum used to control for signal intensity and band shifts. One representative data set from three experiments is shown. Source data are available for this figure.

(CDS, 774 bp) of *O. aries* harboring the VRQ allele (GeneScript), codon optimized for expression in human cell lines and modified to include the human ER localization signal, was cloned into the expression vector (Cat. no. OGS606-5U) under the EF1α promoter (Sigma-Aldrich). Positive clones were verified by Sanger sequencing (Microsynth). After DNA purification, the construct was transfected into cells using Lipofectamine 2000 (Invitrogen). After transfection, cells were kept under antibiotic selection, and single clones were isolated with limiting dilution. In brief, cells were 12× serially diluted 1:2 in a 96-well plate (Merck) starting with 4,000 cells in the first column. Wells with polyclones and single clones were grown to confluency and tested for PrP$^C$ expression. Cells were cultured in OptiMEM (Life Technologies [Gibco]) supplemented with 1% GlutaMAX (GM), 1% MEM Non-Essential Amino Acids (MEM-NEAA; Life Technologies), 1% penicillin/streptomycin (Thermo Fisher Scientific), and 10% FBS (Clontech Laboratories). All cell lines (LN229 [Accession Nr: CRL2611], U251-MG [Accession Nr: CVCL0021], SH-SY5Y wild-type [Accession Nr: CRL2266], and SH-SY5Y$^{\Delta PRNP}$ [Pease et al, 2019]) were cultured in 150-cm$^2$ Corning cell culture flasks (Merck), and counted using trypan blue (Thermo Fisher Scientific) in a TC20 Automated Cell Counter (Bio-Rad Laboratories). Ovinized cells were kept under geneticin (G418 sulfate; Life Technologies) selection at a concentration of 400 μg/ml unless stated otherwise. All cells were grown at 37°C in a 5% CO$_2$ atmosphere.

## Prion infection of cells

300,000 cells were seeded in a six-well plate (Corning). The next day, cells were treated with either 0.25% (wt/vol) PG127 produced in tg338 mice or NBH from C57BL/6J mice in a total culture volume of 1.5 ml. After 3 d, the culture medium was replaced with fresh medium containing geneticin. Cells were lysed at different passages. Cells at passage 8 were used for assessment of the presence of PrP$^{Sc}$, infectivity, and seeding properties. For the dissociation of the cells, StemPro Accutase (Thermo Fisher Scientific) was used.

## Immunoblot analysis

For lysis, cells were washed once with PBS and scraped with lysis buffer (50 mM Tris–HCl, pH 8, 150 mM NaCl, 0.5% sodium deoxycholate, and 0.5% Triton-X 100). Total protein concentrations were determined with a bicinchoninic acid assay according to the manufacturer's instructions (Pierce). PK (Roche AG) digestion was performed at a final concentration of 2.5 μg/ml for cell lysates and 25 μg/ml for brain homogenates for 30 min at 37°C. Pronase E (protease from *Streptomyces griseus* Type XIV; Sigma-Aldrich) was diluted in water to a concentration of 2 mg/ml. Final concentration of pronase E used was 200 μg/ml for brain homogenates and 20 μg/ml for cell lysates. Digestion was performed for 45 min at 37°C and terminated by boiling the samples in LDS (Invitrogen) containing 1 mM DTT (Sigma-Aldrich). Samples were loaded onto a 4–12% gradient gel (Invitrogen) and blotted onto a PVDF or nitrocellulose membrane (Invitrogen). Monoclonal anti-PrP POM1 or POM2 antibody (Polymenidou et al, 2008) was diluted to 300 ng/ml in 1% SureBlock (LuBio Sciences) containing PBS-Tween 20 (PBST; Sigma-Aldrich) and incubated overnight at 4°C. Anti-mouse HRP (Bio-Rad Laboratories) was used as a secondary detection antibody, and

immunoblots were developed with Forte HRP substrate (Millipore). Anti-actin antibody m25 (Merck) was used at a dilution of 1:10,000 in 1% Sureblock containing PBST as a loading control. Imaging was performed on either Fujifilm LAS-3000 (Fujifilm) or Vilber systems.

Aggregate enrichment was performed by centrifugation of a total of 150 μg protein at 20,800g in a tabletop centrifuge (Eppendorf 5417r) for 1 h at 4°C. For the analysis of the supernatant, 20 μl of the sample was isolated and further processed as described above. For the aggregate-enriched fraction, the remaining 20 μl of the total volume were either protease digested or analyzed directly by SDS–PAGE and Western blotting as described above.

## Immunocytochemistry and live imaging

Immunocytochemistry was performed by seeding 20,000 cells on coverslips (Thermo Fisher Scientific) coated with a poly-L-lysine solution (Sigma-Aldrich). Fixation was performed 24 h after seeding with 4% PFA (Roth). Anti-PrP staining was performed with POM1 (3.8 μg/ml) in PBS supplemented with 0.5% BSA for 1 h. As a secondary antibody, goat anti-mouse coupled to Alexa Fluor 488 (Invitrogen) was used at a final dilution of 1:400 in 0.5% BSA in PBS. In addition, DAPI (Sigma-Aldrich) was used at 1 μg/ml during the incubation with the secondary antibody. The cells were washed with 0.5% BSA in PBS between each incubation step for four times. Coverslips were mounted on slides using a mounting solution overnight (Agilent Technologies). Imaging was performed on Leica TCS SP5 (Leica Microsystems). For the comparison of the cell growth of PG127-infected and NBH-treated hovS, 20,000 cells/well were seeded in culture medium in 24-well plates (n = 6 for each condition), imaged, and analyzed with a confluency mask using the Incucyte ZOOM system (Essen Biosciences). A *t* test was performed for the final time point (180 h in culture). Vacuolation was imaged with the Incucyte ZOOM system. Data were depicted with GraphPad Prism 8 (GraphPad Inc.).

## ELISpot for the detection of infected single cells

ELISpot membranes (Millipore) were activated by adding 50 μl filtered ethanol/well and washed twice with 160 μl PBS. Three different dilutions of cells per well (40,000, 4,000 and 400) were spotted onto the membrane and dried with a plate thermomixer (Eppendorf) at 50°C. For the control wells, 40,000 cells were used. After drying, plates were stored at 4°C until further processing. 50 μl of 0.5 μg/ml PK in lysis buffer was added to each well and incubated for 90 min at 37°C. After incubation, vacuum was applied to discard the contents, and wells were washed twice with 160 μl PBS. To stop digestion, 160 μl of 2 mM PMSF (Sigma-Aldrich) diluted in PBS was added to the membrane and incubated at room temperature for 10 min. Tris guanidinium thiocyanate was prepared by diluting 3M guanidinium thiocyanate (Sigma-Aldrich) in 10 mM Tris–HCl, pH 8, and 160 μl/well was added to each membrane. After incubation for 10 min, the supernatant was discarded into 2M NaOH and each membrane was washed seven times with 160 μl PBS and blocked for 1 h with 160 μl SuperBlock (Thermo Fisher Scientific) prepared in MilliQ. Remaining blocking solution was removed under vacuum and 50 μl POM1 was added at a dilution of 1:5,000 in TBST (10 mM

Tris–HCl, pH 8, 150 mM NaCl, and 0.1% [vol/vol] Tween 20) containing 1% (wt/vol) nonfat dry milk for 1 h. The supernatant was discarded into 2M NaOH, and wells were subsequently washed seven times with TBST under vacuum. 50 $\mu$l of anti-IgG1-AP (Southern Biotechnology Associates) was used at a 1:4,500 dilution in TBST-1% (wt/vol) nonfat dry milk and incubated for 1 h. Discarding of the supernatant and washing was performed in the same way as the POM1 antibody. 50 $\mu$l of colorimetric AP dye (Bio-Rad Laboratories) was applied and incubated for 16 min. Membranes were washed twice with water, dried, and stored at –20°C in the dark. Quantification was performed by counting PK-resistant spots. Data were visualized using GraphPad Prism 8 (GraphPad Software, Inc.).

### RT-QuIC assay

The reaction buffer of the RT-QuIC consisted of HaPrP23-231 filtered using 100-kD centrifugal filters (Pall Nanosep OD100C34) at a concentration of 0.1 mg/ml, 1 mM EDTA (Life Technologies), 10 $\mu$M thioflavin T, 170 mM NaCl, and 1× PBS (incl. 130 mM NaCl). Cell lysis was performed in PBS with three freeze–thaw cycles. 2 $\mu$l of the lysates were used at two different dilutions (1:50 and 1:250) to assess seeding activity. The RT-QuIC assay was performed in accordance with the previously established protocols (Atarashi et al, 2011; Frontzek et al, 2016). The plate was loaded into a FLUOstar Omega plate reader (BMG Labtech) and the shaking cycles were set as follows: 7× (90 s shaking; 900 rpm [double orbital]; 30 s rest) and 60 s reading. Reading was carried out with excitation at 450 nm and emission at 480 nm every 15 min. The amplification was performed at 42°C for 105 h. Four replicates per sample were measured.

### PMCA

Cell lysates were prepared in PBS with repeated freeze–thaw cycles as described for the RT-QuIC. Brains from tg110 (bovine PrP expressing mice), tgShpXI (Ovine ARQ variant expressing mice), tg338 (Ovine VRQ variant expressing mice), and tg650 (Methionine 129 human PrP expressing mice) were used to prepare the PMCA substrates (Lacroux et al, 2014). PMCA was performed as previously described (Douet et al, 2017). Briefly, PMCA reactions (50 $\mu$l final volume) were seeded with 5 $\mu$l of sample to be tested. PMCA reactions were then subjected to three amplification rounds each comprising 96 cycles (10 s sonication-14 min and 50 s incubation at 39.5°C) in a Qsonica700. After each round, reaction products (one volume) were mixed with fresh substrate (nine volumes) to seed the following round. The PMCA reaction products were analyzed by Western blot for the presence of PrP$^{Sc}$ (material equivalent to 20 $\mu$l of PMCA product per lane). Each PMCA run included a reference ovine scrapie (PG127) and ovine BSE sample (10% brain homogenate dilution series) as a control for the amplification efficiency. Unseeded controls (two unseeded controls for eight seeded reactions) were also included in each run. PrP$^{Sc}$ extraction and Western blotting was performed as previously described (Huor et al, 2017). Immunodetection was performed using an anti-PrP antibody, Sha31 (1 $\mu$g/ml) (Féraudet et al, 2005), which recognizes the amino acid sequences YEDRYYRE (145-152).

### Mouse bioassay

All animal experiments were performed in compliance with institutional and French national guidelines in accordance with the European Union Directives 86/609/EEC and 2010/63/EU. Experiments were approved by the Committee on the Ethics of Animal Experiments of the author's institutions: INRA Toulouse/ENVT (Permit Number: 01734.01).

Mouse bioassays were carried out in ovine VRQ PrP transgenic mice (tg338), which are considered to be highly efficient for the detection of sheep scrapie infectivity (Le Dur et al, 2005). High sensitivity of tg338 mice for detection of PG127 scrapie isolate was previously reported (Andréoletti et al, 2011). At least six mice were intracerebrally inoculated with each sample (20 $\mu$l) lysed in PBS. No acute adverse effects were observed upon injection of mice. Mice were clinically monitored until the occurrence of TSE clinical signs, upon which time they were euthanized.

## Supplementary Information

## Acknowledgements

A Aguzzi is the recipient of an Advanced Grant of the European Research Council and grants from the Swiss National Research Foundation, the Nomis Foundation, the GELU foundation, the Swiss Personalized Health Network (2017DRI17), and a donation from the estate of Dr. Hans Salvisberg.

### Author Contributions

M Avar: data curation, formal analysis, validation, investigation, visualization, methodology, and writing—original draft, review, and editing.
D Heinzer: data curation, formal analysis, validation, investigation, visualization, methodology, and writing—original draft, review, and editing.
N Steinke: data curation and investigation.
B Doğançay: data curation and formal analysis.
R Moos: data curation and formal analysis.
S Lugan: data curation.
C Cosenza: data curation.
S Hornemann: supervision and writing—original draft, review, and editing.
O Andréoletti: data curation, formal analysis, supervision, investigation, methodology, and writing—original draft.
A Aguzzi: conceptualization, resources, supervision, funding acquisition, visualization, and writing—original draft, review, and editing.

### Conflict of Interest Statement

The authors declare that they have no conflict of interest.

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
