## [Reviewer comments · Life Science Alliance]

Prion infection, transmission and cytopathology modelled in a low-biohazard human cell line

Merve Avar, Daniel Heinzer, Nicolas Steinke, Berre Doğançay, Rita Moos, Severine Lugan, Claudia Cosenza, Simone Hornemann, Olivier Andréoletti and Adriano Aguzzi

DOI: 10.26508/lsa/202000814

Corresponding author(s): Prof. Adriano Aguzzi (University Hospital Zurich)

Review timeline:

Submission Date:	2020-06-16
Editorial Decision:	2020-06-18
Revision Received:	2020-06-19
Accepted:	2020-06-19

Transaction Report:

Please note that the manuscript was reviewed at *Review Commons* and these reports were taken into account in the decision-making process at Life Science alliance.

No Peer Review Process File is available with this article, as the authors have chosen not to make the review process public in this case.

June 18, 2020

RE: Life Science Alliance Manuscript #LSA-2020-00814-T

Prof. Adriano Aguzzi
University Hospital Zurich
Dept. of Pathology
Institute of Neuropathology
Zurich, ZH 8091
Switzerland

Dear Dr. Aguzzi,

Thank you for submitting your revised manuscript entitled "Prion infection, transmission and cytopathology modelled in a low-biohazard human cell line". This manuscript was transferred from another journal, and the review process was carried out at Review Commons.

The reviewers who evaluated your study appreciated your findings, and we would therefore like to publish your work here. You already provided a point-by-point response to the concerns raised by the reviewers, and we have carefully evaluated your response. I think your response addresses the concerns in a good way, and I would thus like to ask you to upload a revised version of your manuscript, including the following changes:

- please fill in all mandatory information in the submission system (i.e. Category, Alternate Abstract, etc.)
- please add a conflict of interest statement to the main manuscript text
- please upload your figure files as separate files while leaving the figure legends in the main manuscript text
- please add a callout for Fig. 3D
- please upload your manuscript in the doc or docx format
- We noticed that your supplementary figures are source data. Thank you for sharing your source data, and we will publish this alongside your paper. However, these should not be part of the main manuscript, so please upload the source data as 'source data files' rather than supplementary figures.

A. FINAL FILES:

B. MANUSCRIPT ORGANIZATION AND FORMATTING:

Sincerely,

Reilly Lorenz
Editorial Office Life Science Alliance

Meyerhofstr. 1
69117 Heidelberg, Germany
t +49 6221 8891 414
e contact@life-science-alliance.org
www.life-science-alliance.org

RE: Life Science Alliance Manuscript #LSA-2020-00814-TR

Prof. Adriano Aguzzi
University Hospital Zurich
Dept. of Pathology
Institute of Neuropathology
Zurich, ZH 8091
Switzerland

Dear Dr. Aguzzi,

Thank you for submitting your Research Article entitled "Prion infection, transmission and cytopathology modelled in a low-biohazard human cell line". It is a pleasure to let you know that your manuscript is now accepted for publication in Life Science Alliance. Congratulations on this interesting work.

DISTRIBUTION OF MATERIALS:

Again, congratulations on a very nice paper. I hope you found the review process to be constructive and are pleased with how the manuscript was handled editorially. We look forward to future exciting submissions from your lab.

Sincerely,

Reilly Lorenz
Editorial Office Life Science Alliance

Meyerhofstr. 1
69117 Heidelberg, Germany
t +49 6221 8891 414
e contact@life-science-alliance.org
www.life-science-alliance.org